# Seed Vigour and Morphological and Physiological Characteristics of *Epimedium brevicornu* Maxim: In Different Stages of Seed Development

**DOI:** 10.3390/plants11182399

**Published:** 2022-09-15

**Authors:** Pengshu Li, Jiarong Fan, Chengli Song, Xuehui Dong, Dingming Kang

**Affiliations:** College of Agronomy and Biotechnology, China Agricultural University, Beijing 100193, China

**Keywords:** *Epimedium brevicornu* Maxim, seed vigour, seed development time, morphological and physiological characteristics

## Abstract

*Epimedium brevicornu* Maxim is a traditional Chinese medicinal plant with important value for curing several diseases, including liver cancer. Seed germination, field seedling emergence, and morphological and physiological traits were measured in developing seeds of *E. brevicornu*, which were collected at 7, 14, 21, 28, and 35 days after flowering. The results showed that with the fruit pericarp changing from lime green to dark red, the seed volume increased. Furthermore, the dry mass of seeds gradually increased from 0.011 g at 7 d to 0.275 g at 35 d, which was a significantly positive correlation with seed vigour (r = 0.980). The soluble protein content initially increased and then decreased to 11.09 mg/g and presented a maximum at 28 d; however, the soluble sugar content gradually declined to a minimum of 30.45 mg/g at 35 d, which was also significantly negatively correlated with seed vigour (r = −0.915). Furthermore, the unsaturated fatty acids (oleic acid and linoleic acid) increase with seed development. Abscisic acid (ABA) reached a maximum value of 18.45 ng/g at 28 d, and gibberellin (GA_3_), 3-Indoleacetic acid (IAA) and zeatin-riboside (ZR) initially increased and then decreased. These results suggest that the vigour of *E. brevicornu* seeds is closely associated with their stage of development, with the highest vigour observed at 28~35 d after flowering.

## 1. Introduction

Vigour is an important indicator of high-quality seeds and an integrative term that includes seed germination, field seedling emergence and seedling resistance under stress [1].

Seed development is closely related to seed vigour, and seeds of most plants have high vigour only after reaching physiological maturity [2]. Reports have indicated that the germination potential of *Gentiana macrophylla* Pall. seeds first increased and then decreased in the different development stages after reaching physiological maturity [3]. Liu et al. [4] found that the germination of the seeds of *Prunella vulgaris* L. reached the maximum value when the fruits became fully yellow after reaching maturity.

The stage of seed development determines the seed size (seed weight) and colour, resulting in significant differences in seed vigour. Su [5] found that *Betula albosinensis* Burkill seed germination was significantly and positively correlated with seed weight. Yang et al. [6] defined the seed coat colour variation of *Paris polyphylla* Smith var. *yunnanensis* seeds as the basis of seed development. They found that when the seed coat colour changed to dark red, seed size, seed weight, germination, and seedling establishment were remarkably higher than seeds of the other two coat colours. Zhang [7] showed that seed vigour was highest when the seed coat of *Platycodon grandiflorum* (Jacq.) A. DC. turned black and had a slender striped texture on its surface. Therefore, seed vigour can be screened by seed coat colour.

During seed development, there is a constant accumulation of different seed storage materials [8], which is also a physiological parameter for a mature seed. Thus, the estimation of a mature seed depends on the colour of the pericarp covering the seeds and the internal materials deposited in a seed [9]. Qi [10] studied the main physiological and biochemical indicators of seeds at different stages of development of *Schisandra sphenanthera* Rehd.et Wils. and found that seed moisture content continuously decreased, but soluble sugar content simultaneously increased, reaching the maximum at 60 d after flowering. Additionally, the soluble protein content peaked at 80 d after flowering, and endogenous hormones showed varied degrees of fluctuation. Wang [11] suggested that the time after flowering can be used as an indicator for harvesting sunflower seeds used for oil and also found that seeds had high linoleic acid and low oleic acid when harvested one week later than the normal mature period. Still, it was beneficial for obtaining high-quality seeds. The endogenous hormone content in seed development of *Paeonia suffruticasa* Andr. fluctuated while the endosperm was filling; however, the soluble sugar content reached a maximum of 50 d after flowering [12].

*E. brevicornu* Maxim, a traditional Chinese herb, is a perennial herb of the genus *Epimedium* Linn. in the family *Berberidaceae*, which is native to Midwest China, such as Shanxi, Gansu, Shanxi, Henan, Qinghai, Hubei, and Sichuan [13]. The seeds of *E. brevicornu* are oblong columnar-reniform and produced in capsules, and they have a caruncle and an appendage structure similar to an air sac [14]. The fruits are prone to dehiscence and shatter during seed maturation in the field; accordingly, it is difficult to harvest and distinguish the seeds in different maturity and vigour levels.

In this paper, the morphological characteristics, soluble sugar, soluble protein, and endogenous hormone contents and vigour of *E. brevicornu* seeds at different stages of development were measured and tested. We established the relationship between the morpho-physiological characteristics of the seed development process and the formation of seed vigour in *E. brevicornu*, which could be a basis for selecting the appropriate harvesting period for high-vigour seeds.

## 2. Results

### 2.1. Seed Morphology of E. brevicornu at Different Stages of Development

The fruits were elongated and soft and had a high moisture content at the beginning of seed formation. As the seed shape changed visibly, the fruits gradually expanded to full size, and the pericarp colour changed from lime green to dark red. The seed colour was greenish. The seed size increased in length and width, and the whole volume gradually increased with time (Figure 1A).

The fresh weight of 100 seeds was 0.129 g at 7 d; the maximum was 0.810 g at 28 d and then declined to 0.713 g at 35 d (Figure 1B). However, the seed moisture content varied based on a continuously decreasing trend with the development stages (Figure 1C). It decreased from 93.25% to 86.36% from 7 to 14 d, in an average decrease of 0.78 %/d. It declined by 0.98 %/d from 14 to 21 d and to 1.34 %/d from 28 to 35 d, which was the maximum.

The dry mass of 100 seeds was only 0.011 g at 7 d, and the dry mass accumulation rate of the seeds accelerated from 7 to 14 d, with an average mass gain rate of 8.00 mg/d. The maximum was from 21 to 28 d; meanwhile, the seed shape was fixed (Figure 1B). The average dry mass gain rate reached a maximum of 18.57 mg/d. Then, the dry mass increase slowed from 28 to 35 d. Furthermore, the average mass gain rate decreased to 6.00 mg/d at 35 d, at which point the seeds were fully mature, and the seed dry mass reached a maximum value of 0.275 g/100 grains. The seed dry mass change curve showed an increase and then a decrease (Figure 1D).

The internal microstructure of *E. brevicornu* seeds is shown in Figure 1E. The endosperm consisted of parenchyma cells, and the seed embryo was extremely small and difficult to be observed at 7 d. However, the primary differentiated proembryonal cells could be observed at 21 d, and HEMs were loose in structure. As the seeds developed, the proembryonal cells could be clearly seen at 28 d and gradually differentiated into a proembryonal cell mass and formed into a globular embryo. The globular embryo was obvious, and the seed morphology was completed at 35 d.

### 2.2. Physiological Characteristics of E. brevicornu Seeds at Different Stages of Development

#### 2.2.1. Soluble Sugar Content of *E. brevicornu* Seeds at Different Stages

Soluble sugars are the basis for seed morphology and material accumulation. As shown in Figure 2A, the soluble sugar content gradually decreased during *E. brevicornu* seed development. After the peak of soluble sugar content appeared at 7 d, the soluble sugar content decreased steadily to 35 d.

#### 2.2.2. Soluble Protein Content of *E. brevicornu* Seeds at Different Stages of Development

The changes in soluble protein content during the development of *E. brevicornu* seeds are shown in Figure 2B. The soluble protein content was stable at 7~14 d, increased sharply at 21 d, remained high until 28 d, and decreased to 9.45 mg/g when the seeds were fully mature by 35 d.

#### 2.2.3. Fatty Acid Content of *E. brevicornu* Seeds at Different Stages of Development

The total fatty acid content of *E. brevicornu* seeds showed a fluctuating trend during seed development (Table 1), which was the same as the fresh weight development of seeds. It reached the maximum of 15.13% at 28 d, which showed that it was an important stage in fatty acid accumulation. Moreover, fatty acid accumulation began at 7~14 d.

*E. brevicornu* seeds are rich in fatty acids. Eleven kinds of fatty acids were detected in our study in general order of linoleic acid > oleic acid > palmitic acid > other fatty acids, in which myristic acid, stearic acid, arachidonic acid, behenic acid, tetradecanoic acid and palmitoleic acid all accounted for less than 0.1%. The unsaturated fatty acids were much higher in the seeds, which also showed linoleic acid > oleic acid > linolenic acid > arachidonic acid > palmitoleic acid, and all of the saturated fatty acids content was lower, in which they showed palmitic acid > stearic acid > arachidonic acid > myristic acid > behenic acid > tetradecanoic acid (Table 1).

#### 2.2.4. Endogenous Hormones in *E. brevicornu* Seeds at Different Stages of Development

As shown in Figure 3A, the abscisic acid (ABA) content in *E. brevicornu* seeds decreased from 14.99 to 12.69 ng/g with seed development, thus showing a trend of first increasing and then decreasing, with a peak of 18.45 ng/g at 28 d. The ABA content rapidly increased to 14.99 ng/g at 14 d and then decreased to 13.83 ng/g at 21 d. Gibberellin (GA_3_) content showed a trend of initially increasing and then decreasing during seed development. It peaked at 41.35 from 7 to 21 d. 3-Indoleacetic acid (IAA) content peaked (0.71 ng/g) at 14 d, which was maintained at a relatively stable level until 28 d.

Zeatin-riboside (ZR) content of *E. brevicornu* seeds largely changed with the stages of seed development, peaking at 71.23 ng/g at 14 d and then decreasing to 33.86 ng/g at 21 d, which had a similar variation with IAA in the whole developing process, with a minimum of 21.33 ng/g at 35 d.

Several ratios of GA_3_/ABA content of *E. brevicornu* seeds are shown in Figure 3B. A rapid high in GA_3_ and a significant elevation in GA_3_/ABA were observed from 1.52 to 2.81 from 7 to 21 d. GA_3_/ABA was decreased from 2.99 at 21 d to 0.57 at 35 d, which could mean ABA was beginning to accumulate and GA_3_ was being consumed during 28~35 d.

The ratio of IAA/ABA during the development of *E. brevicornu* seeds was similar to that of ZR/ABA, thus showing an initial increase and then a decrease. Because the IAA and ZR contents increased significantly from 7~14 d, the ratio of IAA/ABA increased from 0.031 to 0.047, and the ratio of ZR/ABA increased from 4.47 to 4.73. Nevertheless, the ratio of IAA/ABA declined from 0.037 to 0.028, and the ratio of ZR/ABA decreased from 2.42 to 1.85 from 21~35 d.

### 2.3. Vigour of E. brevicornu Seeds at Different Stages of Development

#### 2.3.1. Chamber Germination of *E. brevicornu* Seeds at Different Stages of Development

The germination, sprouting and rotting of *E. brevicornu* seeds at different stages of development with low-temperature stratification of 140 d is shown in Figure 4. The seed germination was 67.02% at 35 d and 59.29% at 28 d, with no significant difference between the two stages. Still, the seed germination at 21 d was 30% significantly lower than at the other two stages. The percentage of rotten seeds was 40.12%, 13.81% and 12.86% at 21, 28, 35 d, respectively (Figure 4).

#### 2.3.2. Seedlings Emergence of *E. brevicornu* Seeds at Different Stages of Development

The seedling growth from *E. brevicornu* seeds at three stages of development is shown in Table 2. Seedlings emerged 7 d after sowing, and the seeds collected 35 d after flowering had significantly higher emergence than seeds from the other two stages of development, with 46.33% after 28 d of sowing. In contrast, the seedling emergence of seeds at the other two stages was less than 20.33%. Seedlings developed the first true leaf at 14 d after sowing, and the percentages of seeds that developed to the first true leaf at the three stages were 49.60%, 59.33%, and 60.09%. The percentages of seedlings with the first true leaves accelerated and reached 67.58%, 77.08% and 90.35% at 21 d after sowing, while the seeds collected 35 d after flowering were significantly higher than those collected 21 d. After 28 d of sowing, the percentage of seedlings that developed the second true leaf was 7.88%, 14.33%, and 42.87%, and the seedling development of seeds collected 35 d after flowering.

### 2.4. Embryo Development of E. brevicornu Seeds at Different Stages of Development

The morphology of *E. brevicornu* seeds changed significantly after 60 d of variable temperature stratification. The seed coat became darker, the seeds became fuller, the abdominal suture line depression was not obvious, and the seed surface became smooth. Although the seed coat became completely black, the radicle of the seed (at 28 and 35 d) broke through the seed coat after 140 d of low-temperature stratification. In comparison, the seed embryos of seeds sampled 21 d after flowering mostly stayed in the late cotyledon type (Figure 5A).

Embryo change of *E. brevicornu* seeds at three stages of development during stratification was investigated (Figure 5A). The embryos of seeds sampled at 21 d did not change significantly relative to that of the variable temperature stratification at 60 d, while the seed embryos of seeds sampled at 28 d developed into the heart type, and that of 35 d seeds developed into the torpedo type. However, the morphology of embryo seeds changed significantly with the low-temperature stratification in 140 d. The radicle of seeds (28 and 35 d) broke through the seed coat, while the embryos of seeds collected 21 d after flowering mostly stayed in the late cotyledon type.

The effect of stratification on the embryos of *E. brevicornu* seeds at each stage of development was significantly different (Figure 5B). The embryo growth of seeds at all three development stages was low. Variable temperature stratification was observed at 60 d, with the largest change of 11% observed for seeds collected at 28 d, followed by 8.3% of 35 d seeds. In contrast, the embryo growth of 21 d seeds was almost 0%. The embryo growth of seeds grew rapidly at all development stages, with low-temperature stratification at 140 d, which was 86.2% at 35 d and 69.1% at 21 d. The result demonstrates that the stratification method was more effective in promoting the physiological maturation of *E. brevicornu* seeds.

### 2.5. Identification of E. brevicornu Seed Vigour by Hyperspectroscopy

Hyperspectral detection of *E. brevicornu* seeds with 21, 28 and 35 d three lots of samples without stratification treatment. There was a significantly different hyperspectral response approximately at 400~1000 nm, and the reflectance was 35 d > 28 d > 21 d approximately at 580 nm (Figure 6). We pre-treated the hyperspectral data at 21, 28 and 35 d with MSC and SNV [15] and then modelled by SVM with a partition of training set: test set: prediction set = 2:1:1. In contrast, the effect of the SVM model with MSC treatment was the best, and the accuracy of the test set was 80%, although the prediction set was up to 95.5% (Table 3).

### 2.6. Association Analysis of E. brevicornu Seed Morphological and Physiological Characteristics with Seed Vigour

A correlation analysis was performed for each vigour index (Table 4). Among them, germination was highly significantly and positively correlated with the hyperspectral reflectance (r = 0.971 **), the hundred fresh weight (r = 0.942 *), hundred dry mass (r = 0.980 **), soluble protein content (r = 0.915 *), and seedling emergence (r = 0.983 **). The dry mass was negatively correlated with the soluble sugar content (r = −0.943 *), but it was positively correlated with the germination (r = 0.980 **) and seedling emergence (r = 0.965 **). Accordingly, the soluble sugar content decreased as the seed vigour level rose with seed development.

## 3. Discussion

### 3.1. Morphology and Physiological Characteristics of E. brevicornu Seeds at Different Stages of Development

Seed coat colour, seed embryo, endosperm and seed morphology, as well as the accumulation of nutrients, such as soluble sugar, soluble protein, starch, and fat contained in seeds, changed dynamically during the formation of seeds [16]. We found that the seed size of *E. brevicornu* increased with development time, reaching a maximum of 35 d after flowering. Still, the moisture content of seeds decreased, reaching a maximum fresh weight at 28 d and a maximum dry mass at 35 d after flowering. Wang [17] found that flowering time varies greatly among four *Epimedium* species, including *E. flavum* Stearn, *E. pubescens* Maxim, *E. chlorandrum* Stearn and *E. davidii* Franch under the same growth environments, even within the same species. However, seed maturation generally occurs approximate 27~36 d after flowering, whereas these plants had the highest fruiting, and the seeds were completely mature. Gao showed that the whole flowering period of *E. pubescens* Maxim was approximate 36 d and the number of flowers was significantly a positive correlation with setting fruit set [18]. In addition, Ying reported that the highest number of fruits and seeds were harvested in *E. wushanense* when the flowering number reached 95% [19]. Thus, the seed development and maturity of *Epimedium* is generally completed approximately 30 d after flowering. However, these stages also depend on several factors, such as the *Epimedium* types, sowing periods, ecological and geographical environments and cultivated techniques.

The metabolic slowdown indicates the completion of seed development, the conversion of soluble proteins to storage proteins with no apparent physiological activity, and the lower saturated fatty acid content and the higher unsaturated fatty acid content due to late dehydration during seed maturation [20,21,22]. In our study, the soluble protein content of *E. brevicornu* seed initially increased and then decreased 28~35 d after flowering, and the unsaturated fatty acid content and fresh weight peaked. Moreover, the soluble sugar content was low, and depletion increased significantly 28~35 d after flowering, whereas the same trend was observed for the starch content. The medicinal plant seeds of *Ginkgo biloba* L. [23], *Panax notoginseng* (Burkill) F. H. Chen ex C. [24], and *Taxus* × *media* Rehder [25], also showed the same trend.

In addition, the ABA content in *E. brevicornu* seeds peaked while the ZR, IAA and GA content decreased at 28 d after flowering. Reports have indicated that ABA in wheat seeds can promote either the synthesis of storage proteins and lipids or germination and synergistically regulates the balance between different hormones by stimulating the rapid synthesis of pre-developmental seed substances and pre-harvest sprouting [26]. We displayed the change in the ABA content in different development stages of *E. brevicornu* seeds; however, we should focus on the concentration change of ABA in the stratification pretreatment of *E. brevicornu* seed germination, which is a necessary step for breaking the dormancy of *E. brevicornu* seeds. Moreover, the ABA function and applied technology in breaking the dormancy of *E. brevicornu* seeds should be further studied in the future.

### 3.2. Association of Development Time and Seed Vigour Formation in E. brevicornu Seed

The seeds of *E. brevicornu* were sound and stable at 35 d after flowering, while their dry mass and moisture content reached 0.28 g and 62.77%, respectively. The ratio of the germinated seeds and seedling emergence at 35 d was highest, the rotten seeds were few among the three stages of seed development (21, 28 and 35 d after flowering), and the ratio of first true leaf and second true leaf were all significantly higher than those of the other two stages. In addition, the germination and seedling emergence at 21 d was significantly lowest among the three stages of seed development due to insufficient maturity and 40% of rotten seeds (Figure 4, Table 2). Yan et al. [2] found that seed germination and seedling emergence in a field of *Platycodon grandiflorum* (Jacq.) A. DC. seeds were consistent with the seed maturity, and the maturity of seeds played a key role in the high vigour of seeds. Chen et al. [27] divided the mature levels of *Alisma orientalis* (Sam.) Juzep. seeds into young, mature and old groups. The results of our study demonstrated that the maturity and vigour of *E. brevicornu* seeds were closely correlated, and seed vigour reached the maximum at 28~35 d after flowering, which can be used as a parameter suitable for harvesting seeds to prevent fruits from cracking.

### 3.3. Hyperspectral Seed Vigour Detection Technique in E. brevicornu

The hyperspectral seed vigour detection technique combines spectral information with image information for nondestructive sample detection of seed vigour. We attempted to detect the vigour of *E. brevicornu* seeds sampled at 21, 28, and 35 d after flowering with the hyperspectral technique and constructed the model with support vector machines (SVM). The result showed that the accuracy of the test set was 80%, and the accuracy of the prediction set could reach 95.5% (Table 3). When Kandpal et al. [28] employed a hyperspectral imaging technique combined with the PLS-DA algorithm to determine the vigour of muskmelon seeds, the accuracy of the test set reached 94.60%. Nansen [29] used hyperspectral imaging combined with a linear discriminant analysis (LDA) algorithm to achieve a 79% test set for vigour level identification of native Australian plant seeds. All data show that the hyperspectral seed testing technique is an efficient methodology for seed vigour via detection nondestructive sampling.

### 3.4. Suitable Harvesting Period for E. brevicornu Seeds

The infinite inflorescences characteristic of *E. brevicornu* makes it difficult to collect seeds in either wildland or fields because the fruits tend to start dry cracking during the ripening period. Accordingly, it is important to identify parameters to determine the suitable harvesting period of the fruits to collect seeds before they dry cracking. Wu [30] reported that the duration of the mature period of *E. brevicornu* seeds generally occurred from 14~20 d, which was mainly influenced by environmental humidity, light illumination and temperature. Tang [31] suggested that 30 d after flowering is the best time to harvest seeds after studying several parameters, including germination percentage, thousand grain weight, soluble sugar content and SOD activity, of infinite inflorescence plants of *Desmodium styracifolium* Merr. Li [32] proposed that the adequate seed harvesting period of *Scutellaria baicalensis* Georgi should be approximate 24~30 d because the thousand seed weight and the vigour index reached the highest value at 30 d. In contrast, the seeds were easily dropped down at 33 d after flowering. The vigour of *E. brevicornu* seed harvested from 28 to 35 d after flowering reached the top point, which was substantiated in the abovementioned publications. Therefore, the several morphological and physiological characteristics we studied could be used as the parameters to estimate the appropriate harvesting period of *E. brevicornu* seeds. We suggest that the optimal harvesting period of *E. brevicornu* seed is 28~30 d after flowering before fruit dry cracking.

## 4. Materials and Methods

### 4.1. Sampling Method

*E. brevicornu* seeds were sampled between July and August 2019 in Zhuoni County, Gannan Tibetan Autonomous Prefecture, Gansu Province. One hundred *E. brevicornu* plants uniform in flowering were selected in the experimental field. Then, we marked every flower with consistent growth stages on each plant and cut off the rest of the flowers and fruits. The sampling stage was 7, 14, 21, 28 and 35 d after flowering (The last sampling was performed at 35 d, as approximately 90% of the fruits in the experimental field were dry cracking at 35 d after flowering). Fifty fruits were selected, with three replications for each sampling. Each fruit contains 3~4 seeds, and the seed samples were immediately placed into liquid nitrogen for storage at −70 °C to test for soluble sugars, soluble proteins and endogenous hormones. The other seeds were used to measure the fresh weight, dry mass and moisture content by 100 seeds, each with three replications.

### 4.2. Measurement of Fresh Weight, Dry Mass and Moisture Content of Seeds

In each of the five development stages, seed samples were weighed (W1) and dried in an oven at 105 ± 2 °C for approximately 10 h until reaching a constant weight, cooled to room temperature and weighed (W2) to calculate the moisture content of the seed samples.
$$\mathrm{Moisture}\;\mathrm{content}\;(\%)=\frac{W1-W2}{W1}\;\times \;100\%$$

### 4.3. Microstructure Observation of Seeds

Paraffin sectioning was used for microstructure observation of seeds [33]. Ten to twenty seeds in each of the five stages of development were immersed in formalin acetate alcohol (FAA) mixed fixative and vacuumed for 6~8 min. Samples were dehydrated in gradient ethanol and xylene, embedded in wax, and then sliced with a Leica RM2256 into 8 μm serial slices. Then the slices were dewaxed in gradient xylenol and ethanol and stained with Safranin O-fast green. Finally, the sections were sealed with Canadian gum and observed and photographed under an Olympus microscope.

### 4.4. Detection of Soluble Sugar Content of E. brevicornu Seeds

The soluble sugar content was measured by the anthrone method [34].

For each of the five development stages of *E. brevicornu* seeds, 0.05 g of ground seeds was placed in a 20 mL tube with 15 mL distilled water. The mixture was placed in a thermostat water bath at 100 °C for 20 min and cooled to room temperature. Then, the samples were filtered into a 100 mL volumetric flask, rinsed several times with distilled water and stored in a refrigerator at 4 °C for backup.

### 4.5. Detection of Soluble Protein Content of E. brevicornu Seeds

Soluble protein content was measured by the Komas Brilliant Blue G-250 method [35].

After weighing 0.05 g of each of the five stages of development of *E. brevicornu* seeds, 5 mL of distilled water was added to a 20 mL mortar. The mixture was ground into a homogenate and then centrifuged at 9000 r/min for 3 min, and the supernatant was set aside. Then, 5 mL Komas Brilliant Blue reagent was added to 1 mL of the soluble protein extract of *E. brevicornu* seeds, and the mixture was shaken well. The absorbance was measured by the colourimetric method at 595 nm for 2 min, which was repeated 3 times.

### 4.6. Analysis of Fatty Acid Content of E. brevicornu Seeds

The fatty acid composition and content were measured by gas chromatography [36].

For fatty acid extraction: seeds samples at five stages of development were dried to constant weight at 45 °C, crushed and passed through a 60-mesh sieve. A sample of 0.1 g was placed into a 15 mL tube and 4 mL of anhydrous methanol: chloroacetyl (*v*/*v*) = 10:1. Mix 5 mL of pentadecanoic acid (C15:0) (≥99%) as the internal standard solution (Sigma-Aldrich, St. Louis, MO, USA) was added. The mixture was shaken well, placed in a thermostat water bath at 80 °C for 2 h and cooled to room temperature. Then, 5 mL of 7% K_2_CO_3_ solution was added and centrifuged at 8000 r/min for 10 min. One millilitre of supernatant was aspirated from the tube and stored in a refrigerator at 4 °C for backup.

Sample measurement and calculation: After the automatic integration of each peak area using an HP7890A gas chromatograph, the content of each fatty acid component was obtained.
$$\mathrm{Fatty}\;\mathrm{acid}\;\mathrm{content}\;(\%)=\frac{{\mathrm{PA}}_{x}\times \left(C_{\mathrm{STD}}\times {\;V}_{\mathrm{STD}}\right)}{{\mathrm{PA}}_{\mathrm{STD}}\times {\;m}_{s}\;}$$

Here, PA_x_ refers to the peak area of fatty acids. PA_STD_ is the peak area of internal standard. C_STD_ is the concentration of internal standard. V_STD_ is the volume of internal standard (mL). m_s_ is the sample mass (g).

### 4.7. Assay of Endogenous Hormone Content of E. brevicornu Seeds

The hormone content was measured by the enzyme-linked immunosorbent assay method [37].

A 0.1 g sample of each of the five development stages of *E. brevicornu* seeds was placed in a precooled (0 °C) mortar. Then, 1 mL of 80% methanol was added, ground into a homogenate and centrifuged at 5000 r/min for 10 min at 4 °C. The supernatant was aspirated from the tube and stored in a refrigerator at −20 °C for backup.

After the hormone concentration (ng/mL) in the samples was obtained by ELISA standard curve, the hormone content was calculated.

### 4.8. Testing of Seed Germination of E. brevicornu Seeds

Before the germination of the *E. brevicornu* seed samples was tested, stratification was performed as a pretreatment of the seed samples [38]. Seed samples were stratified with 1:7 (*v*/*v*) moist sand (80% moisture content) at 10 °C <-> 20 °C (12 h/12 h) temperature without light conditions for variable temperature stratification for 60 d and then transferred to 4 °C without light to continue low-temperature stratification for 140 d.

To keep the moisture content of the sand at approximately 70%, an appropriate amount of water was added every 15 d.

One hundred seeds from each stage of development (21, 28 and 35 d after flowering) were used for testing seed germination. Seed samples were placed in wet sand with 70% moisture content for stratification germination. After 140 d of low-temperature stratification germination, the seed germination and rotten percentages were calculated.

### 4.9. Measurement of Embryo Length of E. brevicornu Seeds

Twenty seeds were randomly selected for sampling after 60 d of variable temperature stratification, followed by 140 d of low-temperature stratification, and then dissected longitudinally. Seed and embryo length was measured using a Moticam 3000 (Xiamen, China), and the embryo length: seed length ratio was calculated using the following formula:Embryo ratio = Embryo length (cm)/Seed length (cm)

### 4.10. Hyperspectral Detection

*E. brevicornu* seeds of 21, 28 and 35 d after flowering were subjected to the hyperspectral imaging system (Spectral Imaging Ltd., Oulu, Finland). The parameters of the hyperspectral system were first set with an electronically controlled platform moving speed of 1.2 m/s, then 90 seeds (embryo side down) were arranged. The extraction of hyperspectral data of seed samples was continued by setting the threshold values using the HIS Analyzer software. The spectral data processing software was MATLAB 6a (The Math Works, MA, USA).

### 4.11. Seedling Emergence

After 60 d of variable temperature stratification followed by 140 d of low-temperature stratification, 100 seeds (21, 28 and 35 d after flowering) with root lengths between 0 and 10 mm were selected as samples. Seeds were sown in seedling trays with a substrate of nutrient soil: vermiculite (*v*/*v*) = 2:1, moisture content of 60% and sowing depth of 0.5 cm. The growth chamber conditions included a temperature of 10~15 °C, air humidity of 50%~75% and light intensity of 2000 lx~4000 lx. Seedling emergence of field and true leaf emergence percentages were observed daily from 1 to 28 d.

### 4.12. Data Statistic Method

Statistical analyses were conducted using SPSS statistical software and service solutions for Windows version 18.0 (SPSS, Chicago, IL, USA). Each treatment was replicated three times. The data were analyzed using an analysis of variance (ANOVA), and the differences between means were tested using Duncan’s multiple range test (*p* < 0.05 *, *p* < 0.01 **).

## 5. Conclusions

*E. brevicornu* seeds at different stages of development have different physiological characteristics, such as coat colour, size, morphology, moisture content, dry mass, soluble sugars, soluble proteins, starch and unsaturated fatty acids. *E. brevicornu* seed completed its vigour development at 35 d after flowering in Gansu province, China. During the seed development, soluble sugar content significantly declined, the starch and unsaturated fatty acid content increased to a peak, and the seed size and dry mass reached the maximum. The fresh mass of the seeds initially increased and then decreased, and the dry mass of the seeds showed a gradually increasing trend. The seed morphology of *E. brevicornu* was completed at 35 d.

The main results show that at 35 d after flowering, the seed vigour of *E. brevicornu* was the highest, the seed morphology completed, and seed germination emergence, first true leaf ratio and second true leaf ratio show satisfactory values. Despite the above results, the suitable harvest time for *E. brevicornu* in northern China should be about 30 d after flowering to avoid loss due to dry cracking fruits.

## Figures and Tables

**Figure 1 plants-11-02399-f001:**
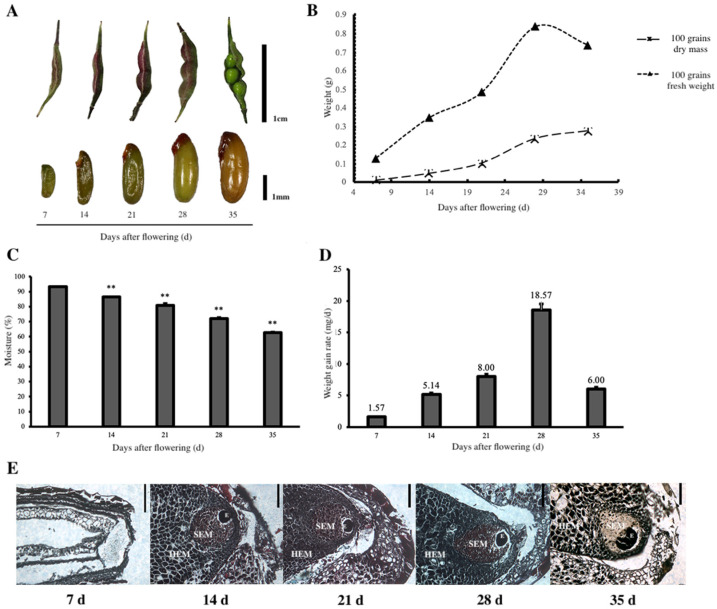
External morphological characteristics and internal microstructure and changes in dry mass and moisture content of *E. brevicornu* seeds at different stages of development: (**A**) Five stages of *E. brevicornu* fruits and seeds, the seeds were sampled at 7, 14, 21, 28, and 35 d after flowering, respectively; (**B**) Changes in fresh weight and dry mass of *E. brevicornu* seeds; (**C**) Changes in moisture content of *E. brevicornu* seeds at different stages of development; (**D**) Changes in the mean mass gain rate of *E. brevicornu* seeds at different stages of development; (**E**) Microscopic observation of *E. brevicornu* seeds in different stages of development. Seed embryo stained with Safranin O -fast green stain; SEM: small endosperm cells; HEM: large endosperm cells; scale bar is 1 mm.

**Figure 2 plants-11-02399-f002:**
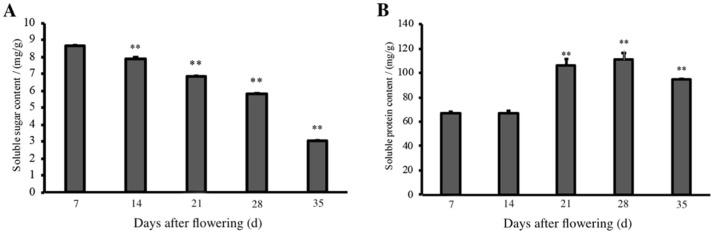
Soluble sugar and soluble protein content of *E. brevicornu* seeds at different stages of development: (**A**) Changes in soluble sugar content of *E. brevicornu* seeds at different stages of development; (**B)** Changes in soluble protein content of *E. brevicornu* seeds at different stages of development. *p* < 0.01 **.

**Figure 3 plants-11-02399-f003:**
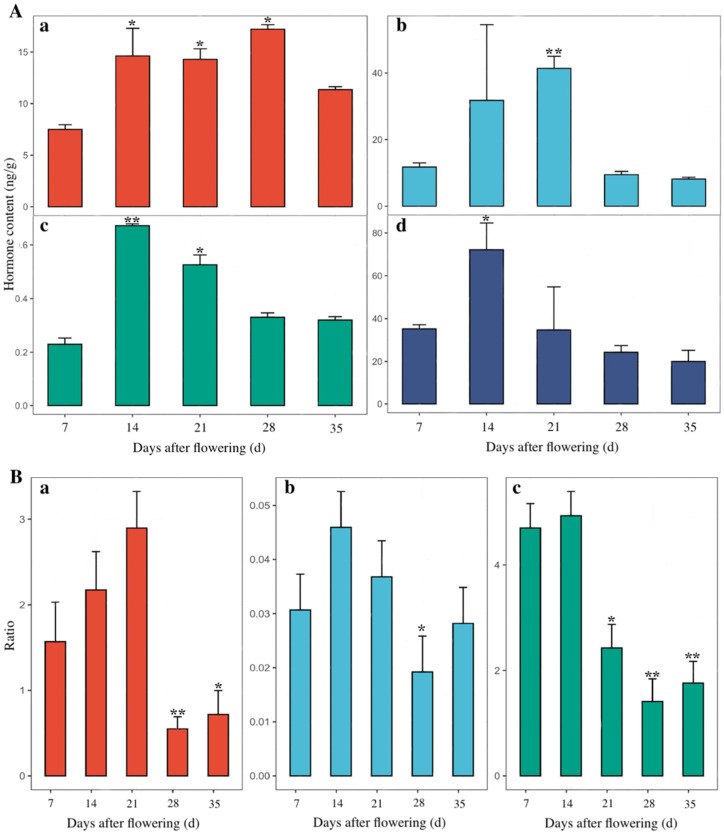
Changes in endogenous hormones contents of *E. brevicornu* seeds at different stages of development. (**A**) Changes in endogenous hormone contents of *E. brevicornu* seeds at different stages of development. (**a**), ABA content; (**b**), GA_3_ content; (**c**), IAA content; (**d**), ZR content. (**B**) Changes in endogenous hormone content ratio of *E. brevicornu* seeds at different stages of development. (**a**), GA_3_/ABA; (**b**), IAA/ABA; (**c**), ZR/ABA. *p* < 0.05 *, *p* < 0.01 **.

**Figure 4 plants-11-02399-f004:**
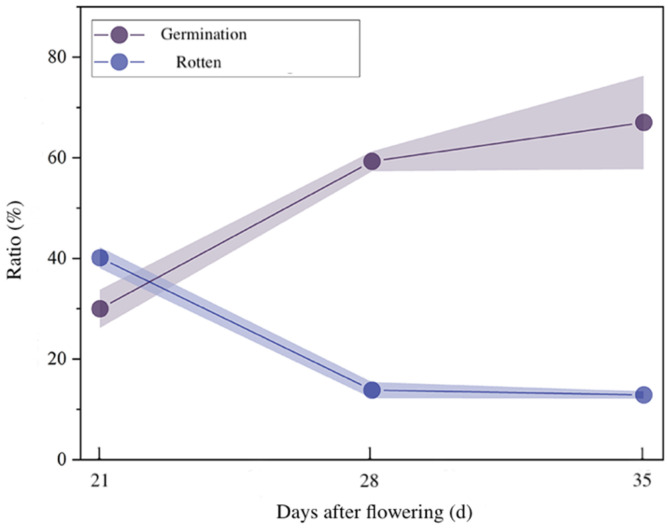
Seed germination of *E. brevicornu* at different stages of development (21, 28, 35 d after flowering). The real line is the average percentage of the germinated or rotten seeds to the total testing seeds in the different stages of development. The shade surrounding the real line is the variation standard error of the percentage of the germinated and rotten seeds in the different stages of development.

**Figure 5 plants-11-02399-f005:**
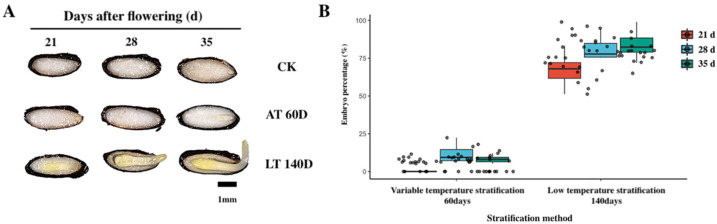
Morphological changes during the stratification of *E. brevicornu* seeds at different stages of development (21, 28, 35 d after flowering): (**A**) Morphological changes of *E. brevicornu* seeds during stratification at different stages of development. CK, seed morphology before stratification; AT60D, seed morphology after 60 d of variable temperature stratification; LT140D, seed morphology after 140 d of variable temperature stratification followed by low-temperature stratification; (**B**) Changes in seed embryo size and morphology of *E. brevicornu* seeds at different stages of development during stratification.

**Figure 6 plants-11-02399-f006:**
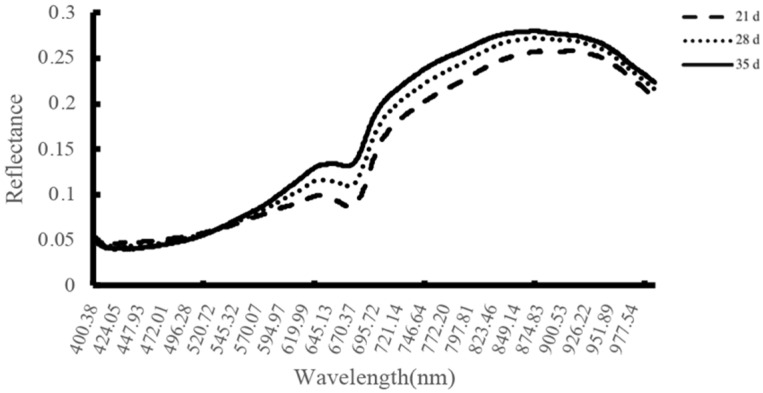
Spectral curves of *E. brevicornu* seeds at different stages of development (21, 28, 35 d after flowering).

**Table 1 plants-11-02399-t001:** The percentage of fatty acid content to dry mass of *E. brevicornu* seeds at different developmental stages (%).

	Days after Flowering (d)	7	14	21	28	35
Saturated fatty acid	Palmitic acid	1.33	1.49 ± 0.01	1.31 ± 0.01	1.67 ± 0.03	1.40 ± 0.01
	Myristic acid	0.05	0.05	0.04	0.05	0.04
	Stearic acid	0.30	0.36	0.31	0.36 ± 0.01	0.29
	Arachidic acid	0.06	0.05	0.04	0.04	0.04
	Behenic acid	0.03	0.02	0.02	0.01	0.01
	Tetradecanoic acid	0.01	0.00	0.01	0.02	0.02
	Subtotal	1.77	1.98	1.73	2.15	1.80
Unsaturated fatty acid	Linoleic acid	6.13	7.17 ± 0.03	6.47 ± 0.03	8.31 ± 0.15	7.02 ± 0.04
	Oleic acid	2.40	3.11 ± 0.01	2.93 ± 0.02	4.41 ± 0.07	3.92 ± 0.01
	Linolenic acid	0.28	0.25	0.19	0.19	0.15
	Arachidonic acid	0.03	0.04	0.03	0.04	0.04
	Palmitoleic acid	0.02	0.03	0.02	0.03	0.03
	Subtotal	8.87	10.60	9.64	12.98	11.16

* Total fatty acids include subtotal saturated fatty acids and subtotal unsaturated fatty acids.

**Table 2 plants-11-02399-t002:** Growth from seedlings of *E. brevicornu* seeds (after 140 d of low-temperature treatment) at different sown days at different stages of development (21, 28, 35 d after flowering).

Growth from Seedlings (%)	Days after Flowering (d)	Days after Sowing (d)			
		7	14	21	28
Emergence	21	4.67 ± 2.33 a	7.67 ± 2.96 a	8.67 ± 3.53 a	9.33 ± 3.84 a
	28	8.00 ± 3.06 a	17.00 ± 2.00 a	19.00 ± 1.53 ab	20.33 ± 1.20 ab
	35	21.33 ± 8.35 b	37.33 ± 7.13 b	46.00 ± 13.20 b	46.33 ± 13.10 b
First true leaf	21		49.60 ± 2.28 a	67.58 ± 9.04 a	75.05 ± 12.96 a
	28		59.33 ± 7.55 a	77.08 ± 0.47 ab	80.71 ± 5.07 a
	35		60.09 ± 6.85 a	90.35 ± 3.56 b	93.20 ± 3.65 a
Second true leaf	21				7.88 ± 1.21 a
	28			4.55	14.33 ± 5.28 a
	35			10.56 ± 2.96	42.87 ± 0.99 b

Note: Means of different treatments followed by the same lowercase letters are not significantly different at the 5% probability level, according to Duncan’s multiple range test.

**Table 3 plants-11-02399-t003:** Results of SVM models based on different pretreatment.

Pretreatment	Model	Training Set	Testing Set	Hold-Out Set
MSC	SVM	88.3	80.0	95.5
RAW		95.0	66.7	95.5
SNV		95.0	73.3	68.2

Note: MSC, multiplicative scatter correction; RAW, no pretreatment; SNV, standard normal variate; SVM, support vector machines.

**Table 4 plants-11-02399-t004:** Correlation coefficients between seed morphological and physiological characteristics with seed vigour.

	RF	FW	DW	SS	TP	FA	GR	ER
RF	1							
FW	0.916 *	1						
DW	0.912 *	0.948 *	1					
SS	−0.855	−0.823	−0.943 *	1				
TP	0.911 *	0.824	0.705	−0.574	1			
FA	0.585	0.852	0.743	−0.511	0.511	1		
GR	0.971 **	0.942 *	0.980 **	−0.915 *	0.811	0.674	1	
ER	0.946 *	0.962 **	0.965 **	−0.839	0.830	0.772	0.983 **	1

Note: RF represents reflectance. FW represent 100 grains of fresh weight. DW represent 100 grains of dry mass. SS represent soluble sugar content. TP represent total protein content. FA represent total fatty acid content. GR represent germination percentage. ER represent emergence percentage. * *p* < 0.05 ** *p* < 0.01.

## Data Availability

Data is contained within the article.

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
