# Peer review of "Seed Vigour and Morphological and Physiological Characteristics of Epimedium brevicornu Maxim: In Different Stages of Seed Development"

_plants, 2022, doi:10.3390/plants11182399_

Round 1

Reviewer 1 Report (Previous Reviewer 3)

Some of the changes in the manuscript are very good, but other changes are not good – in some cases the changes are not correct. Please followed the suggestions to improve the English very carefully.

Line 15. Change “swelled” to “increased”

Line 20.  Change “And” to “Furthermore”   It is not proper English to begin sentences with words such as “and” and “but”; these words are conjunctions.

Line 21. Change “raised” to “increases”

Line 39. Change “displayed” to “reached”

Lines 58 and 59.  Delete the newly-inserted “the”

Line 65.  Delete “well”   Your changes to this line are very unclear.  The original wording was more clear than what you have now. 

Line 73. Change “show” to “have”

Line 76. Change “maturities” to “maturity”

Line 91.  Delete “respectively”

Line 92. Insert “was”  before “performed”

Line 92. Change “approximate” to “approximately”

Line 92.  Insert “of the” before “fruits”

Line 97. Insert “for” before “soluble”

Line 103. Change “approximate” to “approximately”

Line 112.  Not clear what you mean by “were waxed”   Do you mean they slides were dewaxed? On line 111, you say that they were embedded in wax.

Line 126.  Change “was” to “were”            5 milliliters were   --  not was

Line 128. Change “was added” to “were added”

Line 141. Change “was” to “were”

 Line 167. Change “approximate” to “approximately”

Line 172.  Change “percentage” to “percentages”

Line 178. Change wording to “the embryo length : seed length ratio was calculated using the …..”

Line 182. Change “seed” to “seeds”         seeds were

Line 189.  Change the title to “ Seedling emergence”     According to information in the paragraph, the seeds were not planted in the field. The seeds were planted in a growth chamber.

Line 211.  Delete “course”   

Lines 213-217.  All this is a hard-to-follow  run-on sentence that needs to be divided.  Stop on line 216.  “decrease of 0.78%. It declined by 0.98%...”

Line 217. Insert “and”  after “21 d”

Line 221.  Delete “which time”

Line 223.  Change “After the highest point”  to “Then,”

Lines 253.  Change to “sugar content decreased steadily to 35 d.”

Lines 253-254.  The words in red do not make any sense (not clear what you want to say).  I suggest you delete all of the words in red on these two lines  -- not needed.  

Line 265. Change “is” to “are”     seeds are

Line 270.  change “as” to “the”  the meaning of “as” is not clear

Line 277. Change “contain” to “are”

Line 294.  Hard to understand   Change to “peaked (0.71ng.g) at 14 d,”

Lines 298-299.  Delete “of seeds”      otherwise this is hard to read

Line 309.  Change “both of”  to “both the”

Lines 326-327.  I cannot follow/understand these lines.   I think what you are trying to say is this:

    “..other two stages. The percentage of rotten seeds was 40.12, 13.81 and 12.86 at 21, 28 and 35 d , respectively.”    [[Is this correct?]]

Line 339.  You did not sow seeds in the field.  Thus, this line needs to be changed.  Change to “Seedling emergence of E. brevivornu sseeds….”

Lines 335 and 341. Change “growth of” to “ growth from”

Line 347.  A growth rate cannot be a percentage.   If you are talking about “rate” that means you are talking about the speed of leaf growth.   Thus, there is something wrong with this sentence.  Do you mean the percentages of seedlings with the first leaf?  Need to correct this sentence.

Line 352. Delete “showed an absolute advantage”  The meaning of this phrase is not clear – these words are not needed.

Line 376.  “peaked” is not the correct word  to use here.  I suggest you change “peaked” to “collected”

Line 378.  Delete “rate”   you did not measure how fast the embryos grew

Line 380.  Do you mean “picked”

Lines 381, 382. Delete the word “rate”  When you use “rate” then the sentence does not make any sense. 

Lines 387-388.  Need to insert a verb  -  this is not a complete sentence

Line 390.  The meaning of word “rate” is not clear.  Here, do you mean the speed (if so rate is OK),  however, if you mean the % or the amount, then please change the sentence to make it clear.

Line 391.  Need to say how you “pre-treated” the data. 

Line 409. Delete “in the field”  You put the seeds in a growth chamber – not the field.

Line 411. Either delete the comma before “but”  or insert “it” after “but”  The sentence is not correct the way is it now.

Line 434.  Change “rate” to “percentage”  or just simply delete “rate”     Rate means the speed  - not the amount

Line 435. Insert “that” after “showed”

Line 453. Insert “same” before “trend”  

Lines 468, 472.  Delete “of field”

Line 468. Change “were” to “was” 

Line 473.  Change wording to “maturity, and 40% of the seeds rotted….”

Line 480.  Delete “either”  “its”  and “had”

Line 488. Change  “resulted” to “result”     and insert a comma after “80%”

Line 496. Insert “via” after “detection”

Line 498. Insert “to” before “collect”

Line 499. Insert “start” before ‘dry”

Line 514. Change wording to “characteristics we studied could be used…”

Line 524. Delete “its”

 Line 526. Change “weight” to “mass”

Line 532. Change wording to “rotten seeds the lowest; however, during seedling emergence first …”

Author Response

Reviewer 2 Report (Previous Reviewer 2)

In my opinion, the revised version of the manuscript may be accepted for publication. 

Author Response

Reviewer 3 Report (Previous Reviewer 1)

The latest version of the revised manuscript shows a significant improvement in form and presentation. However, the scientific limit in the methodology remains: the seed vigour (size, seed dry weight, seed emergence…) reached the maximum value at 35 days after flowering, which corresponds to the last sampling date carried out by the authors; but these values would have been the same or increased in subsequent sampling date?

For the above reason, I suggest deleting the phrase "We suggest ..." (lines 515-517) in apparent conflict with what  stated in the conclusions at line 537. Moreover, in the conclusions section, delete lines 530-538 and replace it with a sentence like the following:

The main results show that at 35 days after flowering the seed vigour of E. brevicornu was the highest, the seed morphology completed and seed germination emergence of field, first true leaf ratio and second true leaf ratio show satisfactory values. Despite the above results, the suitable harvest time for E. brevicornu in the northern of China should be about 30 days after flowering, in order to avoid loss due to dry cracking fruits.

Author Response

This manuscript is a resubmission of an earlier submission. The following is a list of the peer review reports and author responses from that submission.

Round 1

Reviewer 1 Report

It is mandatory that the English of the entire manuscript must be completely revised by a native speaker before submitting the article for a possible publication.

Line 76-77: the physiological maturation stage of the seed does not increase. It is a very specific and determined stage; once reached it cannot increase. Revise the sentence

Line 195: “in batches”? please explain

Line 257: “with consistent flowering growth”? please explain

Line 258:” 50 pods were taken”, randomly chosen, per plant? Please specify

Line 258-259:  three replicates means 150 pods in total?

Lines 258; 260-261: “at 7 d intervals”? after flowering, please specify.

Lines 264-268: revise the sentence (the word “determined” repeated three times in three lines!).

Line 264: “by 105 °C drying method..” How long?

Lines 268-286. Section 2.3. incomprehensible. please have a native English speaker proofread

Lines 294-299. verbs conjugation? standardize throughout the manuscript

Lines 460-479:  an English native speaker is required to rewrite the text!!

Lines 466-467: “(wet sand is a ball in the hand and falls apart on the ground)”. Scientifically inappropriate sentence.

 Line 464: specify the duration of both stratification treatments.

Line 469: absorb excess water, ………? Incomprehensible sentence

Line 472. Temperature lamination? Please explain this sentence.

Lines 473-474: the sentence” …every 15 d to replace the wet sand with 70% water content, and appropriate supplemental water to keep the sand moisture content at about 70%.” should be replaced with  “…to keep the water content of the sand around 70%, an appropriate amount of water was added every 15 days”.

Lines 477-478: the sentence” … and the wet sand with 70% water content was replaced every 15 d at 15 d intervals, and the water content of the sand was kept at about 70% with appropriate supplementation“ should be replaced with  “…to keep the water content of the sand around 70%, an appropriate amount of water was added every 15 days”.

Lines 484-486:  rephrase the sentence; it is not clear the difference between "the longest distance between the longitudinal poles of the seed profile "and" the vertical distance between the longitudinal poles of the seed profile "

Lines 486-487: specify the number of seeds used and rephrase the sentence as follow:   10? seeds were sampled after 140 d of low temperature stratification treatment, and…….

Line 608: …”a water content of 100-125%..::” please explain and detail this sentence.

Reviewer 2 Report

The study was focused on seed vigor and morphological and biochemical characteristics of Epimedium brevicornu Maxim. in different seed development phases. The results evidenced that the seed pericarp was changing from lime green to dark red and the seed volume was increasing, furthermore, the dry weight of seeds gradually increased from 0.011g at 7 d to 0.275 g at 35 d, which was a significantly positive correlation with the seed vigor (r = 0.980). The maturity and vigor of E.brevicornu seeds were closely related to the phase of seed development, and the seeds reached their highest vigor at 28-35 d after flowering and could be harvested as seed for breeding to prevent pod bursting and harvesting difficulties.

In my opinion, the paper is scientifically interesting and well prepared. However, I recommend few improvements:

  • The Introduction is too long, it may be presented in more concise form,
  • Moderate English changes in the manuscript by the native speaker are required,
  • In Materials and Methods, a separate passage regarding statistical tests should be added,
  • Statistical test results should be presented in figures, for example in Figure 3,
  • Conclusions should be re-written, they should not be a repetition of the results,
  • Some of the references should be replaced with the relevant newer citations.

Reviewer 3 Report

I have written numerous suggestions for word changes on the pages of the manuscript.  Please see attached file.  In general, the English writing is not what you would like for it to be for a scientific paper.

General comments.

“in different seed developing times”  does not sound very good.  I suggest you change all of them to “different stages of seed development”  including in the title and numerous places in the text

The abstract does not mention the size of the embryo or that cold stratification is required for germination

Line 37 and in many places in the text.  The word “rate” means the speed at which something happens.  It should not be used when you are talking about percentage data.

Line 38.  Cob is a word used to describe the infructescence of Zea mays.  I suggest you say “fruit”. 

Line 88-89.  Not clear.  Do you mean that you had three replicates of 50 fruits each?  Why not say that “we collected three replicates of 50 fruits for each stage of development”

Lines 93-94.  Do you mean you had three replicates of 100 each?  Not clearly worded.

Lines 101-157.  In general, the methods do not read well.  They are choppy and often the words are not in good sentences.  Also, there are some run-on sentences, i.e. sentences with three clauses – should only have two clauses per sentence.

Lines 323-329.  It is not clear to me when the embryos grew.  Need to tell the reader if the embryos grew during cold stratification, or if embryo growth was delayed until cold stratified seeds were moved to a higher temperature.

I have written numerous suggestions for word changes on the pages of the manuscript.  Please see attached file.  In general, the English writing is not what you would like for it to be for a scientific paper.

General comments.

“in different seed developing times”  does not sound very good.  I suggest you change all of them to “different stages of seed development”  including in the title and numerous places in the text

The abstract does not mention the size of the embryo or that cold stratification is required for germination

Line 37 and in many places in the text.  The word “rate” means the speed at which something happens.  It should not be used when you are talking about percentage data.

Line 38.  Cob is a word used to describe the infructescence of Zea mays.  I suggest you say “fruit”. 

Line 88-89.  Not clear.  Do you mean that you had three replicates of 50 fruits each?  Why not say that “we collected three replicates of 50 fruits for each stage of development”

Lines 93-94.  Do you mean you had three replicates of 100 each?  Not clearly worded.

Lines 101-157.  In general, the methods do not read well.  They are choppy and often the words are not in good sentences.  Also, there are some run-on sentences, i.e. sentences with three clauses – should only have two clauses per sentence.

Lines 323-329.  It is not clear to me when the embryos grew.  Need to tell the reader if the embryos grew during cold stratification, or if embryo growth was delayed until cold stratified seeds were moved to a higher temperature.

Round 2

Reviewer 1 Report

The authors, despite the large amount of work carried out, come to conclusions either obvious, partial and of very little interest to the reader (see below the comment at the conclusions section).  The text must be thoroughly revised by a native English speaker

Line 40: cob maturity instead of “mature of the cob”.

Lines 41 and 46: erase “thousand” before “seed weight”.

Line 48: erase “outer” before “coat”.

Line 51: different instead of “a varieties of”.

Lines 58 and 59: 60 d and 80 d - for the first time in the text, it is necessary to explain the meaning of d.

Line 60: Wang [11] does not correspond to the reference list.

Line 90-92: The sampling was carried out with an interval of 7 days starting from 7, 14, 21, 28 and 35 days after flowering. For each sampling period, three replicates of 50 pods each were collected. instead of “ The sampling periods……..at 7 days intervals.” 

Line 95: Authors should provide information on the average number of seeds per pod.

Lines 105, 112, 120, 129, 143, 150…: I suggest replacing "developing times" with sampling periods or modify what previously reported.

Line 105: “FFA” - write in full

Line 129 and 152: colon after “Fatty acid extraction” and “Seed stratification method” instead of comma

Line 145: “4°C”; probably at 4°C?

Lines 147-148: revise the sentence.

Lines 150-151 and 168-169: “100 seeds for each of the 3 replications for three different……..were used”, instead of the sentence in the text.

Line 166: “embryo ratio of seed was calculated,” explain how!

Line 174: “A sample of E. brevicornu seed…” explain at what stage of development the seed sample was collected.

Line 186 and 301: growth or germination chamber instead of “culture room”.

Line 200: and instead of “or”, may be.

Line 203 and 215: “a Bell type trend” and “S type growing trend”… please explain.

Lines 227-228: endosperm instead of “edosperm” and review the sentence, please.

Lines 258-259: “arachidonic acid” is repeated twice.

Line 265- Table 1 doesn’t represent the trend of fatty acid composition.

Figure 2 C and D. The figures are not mentioned in the text. Moreover, the figure 2 C has to be redrawn. The maximum accumulation of fatty acid composition, correspondent to the grey column (specify % of what), should have the maximum value on the Y axis and within this column, the relative fractions of unsaturated and saturated fatty acid composition. Figure 2d is illegible

Lines 272-273: 28 d instead of “35 d”; 21 d instead of ”28 d”, please check!

Figure 4- The figure is not mentioned in the text. Moreover, explain the presence of halos on the lines in the figure and insert the bars to indicate standard error of the mean.

Table 1: Insert lines inside the table to separate columns and rows properly. Specify in the caption that the seeds come from the 140d treatment.

Line 329: Erase “We found that”

Figure 5 B: Alternating instead of “Altering”?.  Authors should explain what embrio rate expressed as % means (in materials and methods).

Line 347: 28 d instead of “21 d” and 35 d instead of “28 d”?

Line 352: developing instead of “develoing”

Line 375: the meaning of RAW is missing.

Table 3: Replace the numbers with abbreviations of the name of the characteristics analysed, reported under the table. The interpretation of the table is not immediate. Authors must explain what they mean by seed vigour index.

Discussion section

Lines 400-413: the authors first argue that the flowering time, seed set and seed maturation are highly variable and scalar in the genus Epimendium, depending on the species, environment, cultivation technique, sowing period and then incomprehensibly conclude that two phenological phases, completely distinct and different from each other such as the development of the seed and the maturity occur about 30 days after flowering in the genus Epimendium.  Authors perhaps try to justify the seed collection at the three development times (21, 28 and 35d after flowering) carried out. This choice, in fact, appears arbitrary and the question is: would the conclusions have been the same if others seed samples had been collected later (for example 42 and 50 d after flowering) than the last one (35d after flowering) carried out by the authors? See further comments on this below.

Lines 407-409: flowers instead of “flowering”?

Line 404, 408-409; revise the English

Lines 416-417: The authors should better explain the relationship between the late dehydration during seed maturation and fatty acid profile.

Lines 417-422: The sentences are a repetition of the results obtained

Lines 425-434: No mention about the fundamental role that ABA plays in inducing the primary dormancy of which E. brevicornu seeds seem to be affected.

Line 427: and instead of “or”?

Lines 436-444: The sentences are a repetition of the obtained results.

Lines 440 and 442: and instead of “but”?

Line 444: Table 2?

Line 457: Revise the English form

Line 459: Table 4 is not present in the text.

Lines 463-464: Revise the English form

Lines 474-475: “120d”, may be 140d. Moreover, authors to make this sentence, should have compared the results obtained with seeds subjected to 140 days at low temperature (as carried out) with seeds without any treatment, utilized as control.

Conclusions section

In summary, the authors conclude that the optimal time for collecting and obtaining an E. brevocornu seed with the highest vigour is 35 days after flowering, but this statement, as formulated, cannot be scientifically accepted.  In fact, in order to affirm the above, the authors would have collected further seed samples at a more advanced developing stage (for example at 42 and 50 d after flowering) and verify that the seed of these samples showed the same or worse qualitative characteristics compared to the seed collected at 35 d after flowering. But this was not done, so the authors can only affirm that: “the seed of E. brevicornu shows a higher vigour with respect to the others developmental stages considered, if collected at 35 d after flowering”; significantly reducing the interest of this experimentation.

Lines 496-498: This statement is obvious and valid for all dormant seeds.

The unit of measure present in the title of the axes of all the graphs must be enclosed by round brackets and the slash, where present, must be eliminated.

Reviewer 3 Report

Line 12. Change “development” to “developing”

Line 23. Change “by to “of”

Line 26. Change “or” to “of”

Line 40, Delete “reached”

Line 43. Change “fruit” to “fruits”

Lines 48-52.  This is a run-on sentence.  This needs to be divided into at least two sentences.  A sentence should not have more than two clauses.

Line 79.  Change to “fruits are”   A pod is another word for fruit.  Thus, you are saying fruit fruit.

Line 80. Change wording to “…accordingly it is difficult to harvest seeds and distinguish…”

Line 86, change “relativeness” to “relationship”

Line 97. Change “trial” to “study”

Line 93. Change “was” to “were”

Line 99.  Change to “50 fruits” and change to “each fruit contains”

Line 102. Change “fruit pods” to “fruits”

Line 104. Insert “used” after “were”

Line 106. Insert “seeds” before “each”

Line l09. Change ‘sample” to “samples”

Line 115. Change “section” to “sectioning”

 Line 116. Change “stage” top “stages”

Line 118. Insert “Samples were” before “dehydrated”

Lines 125-126. Change wording to “seeds, 0.05 g. of ground seeds were placed in a 20 ml tube with 15 ml distilled water”

Line 137. Delete “and add” at the first on this line

Line 145. Delete “add”

Line 154. Change “were” to “was”

Line 186. Change “rate” to “percentage”

Line 213. Insert “Seeds were” before “sown”

Line 224. Delete “seed” before “stages”

Lines 225, 227, 253, 531, 558, 575, 595.  Change “fruit pods” to “fruits”

Lines 242-246.  Need to divide into two sentences.

Lines 261, 361. Change “was” to “is”

Fig 2.  Need to remove “C and D” pictures from the figure – you removed the caption

Line 307. Change “rich of fatty acids” to “rich in fatty acids”

Line 382. Change “are” to “is”

Line 398. Delete “more”

Line 415. Change “didn’t” to “did not”

Line 469. Delete “are”

Line 470. Change “increasing’ to “increased”

Lines 473, 483 485.  The word “Epimedium” should be in italics – genus name.   Epimedium

Line 477. Change “mature” to “maturation”

Line 497. Insert “same” before “trend”                I think this  is what you mean.

Lines 517-524.  This is another run-on sentence – too many clauses in it.  Need to divide and make sure each new sentence has only two clauses.

Line 565.  Change “rate” to “percentage”      I strongly doubt that the author recorded only the speed of germination.  Perhaps he/she recorded percentage and rate, but I doubt that this author only recorded speed.  Remember rate = speed of an event.

Line 571. Delete “besides”

Line 573. Insert “used” before “as”

Line 584. Change “was developed to” to “reached”
